# Impacts of Glucose-Dependent Insulinotropic Polypeptide on Orthodontic Tooth Movement-Induced Bone Remodeling

**DOI:** 10.3390/ijms23168922

**Published:** 2022-08-10

**Authors:** Taisuke Yamauchi, Megumi Miyabe, Nobuhisa Nakamura, Mizuho Ito, Takeo Sekiya, Saki Kanada, Rina Hoshino, Tatsuaki Matsubara, Ken Miyazawa, Shigemi Goto, Keiko Naruse

**Affiliations:** 1Department of Orthodontics, School of Dentistry, Aichi Gakuin University, Suemori-dori, Chikusa-ku, Nagoya 4648651, Japan; 2Department of Internal Medicine, School of Dentistry, Aichi Gakuin University, Suemori-dori, Chikusa-ku, Nagoya 4648651, Japan; 3The Graduate Center of Human Sciences, Aichi Mizuho College, Syunko-cho, Mizuho-ku, Nagoya 4670867, Japan

**Keywords:** glucose-dependent insulinotropic polypeptide, orthodontal tooth movement, bone remodeling

## Abstract

Glucose-dependent insulinotropic polypeptide (GIP) exerts extra-pancreatic effects via the GIP receptor (GIPR). Herein, we investigated the effects of GIP on force-induced bone remodeling by orthodontic tooth movement using a closed-coil spring in GIPR-lacking mice (GIPRKO) and wild-type mice (WT). Orthodontic tooth movements were performed by attaching a 10-gf nickel titanium closed-coil spring between the maxillary incisors and the left first molar. Two weeks after orthodontic tooth movement, the distance of tooth movement by coil load was significantly increased in GIPRKO by 2.0-fold compared with that in the WT. The alveolar bone in the inter-root septum from the root bifurcation to the apex of M1 decreased in both the GIPRKO and WT following orthodontic tooth movement, which was significantly lower in the GIPRKO than in the WT. The GIPRKO exhibited a significantly decreased number of trabeculae and increased trabecular separation by orthodontic tooth movement compared with the corresponding changes in the WT. Histological analyses revealed a decreased number of steady-state osteoblasts in the GIPRKO. The orthodontic tooth movement induced bone remodeling, which was demonstrated by an increase in osteoblasts and osteoclasts around the forced tooth in the WT. The GIPRKO exhibited no increase in the number of osteoblasts; however, the number of osteoclasts on the coil-loaded side was significantly increased in the GIPRKO compared with in the WT. In conclusion, our results demonstrate the impacts of GIP on the dynamics of bone remodeling. We revealed that GIP exhibits the formation of osteoblasts and the suppression of osteoclasts in force-induced bone remodeling.

## 1. Introduction

During orthodontic treatment, bone remodeling occurs when orthodontic force is applied to the teeth. Osteoclasts and osteoblasts mediate bone resorption and formation, respectively [1]. In the bone remodeling induced by orthodontic forces, bone resorption is induced by osteoclasts on the compression side and bone formation by osteoblasts on the traction side [2], and the surrounding tissue undergoes constant modifications.

In recent years, the number of middle-aged and elderly patients who desire orthodontic treatment has increased. Adult and elderly patients often have systemic diseases such as diabetes and obesity. Changes in metabolic states that impede bone remodeling can result in variations in tooth movement rates [3,4,5,6]. Diabetes is spreading and is increasing worldwide. In 2019, 463 million people were living with diabetes and the number of cases is expected to increase to 578 million by 2030 [7]. Diabetes is associated with many musculoskeletal complications, including an increased incidence of fractures and osteoarthritis leading to joint pain and loss of function [4,5,8,9,10]. These issues indicate that the improvement of bone remodeling is critical, not only for orthodontic treatment in adult patients but also for the quality of life of patients with diseases such as diabetes.

Glucose-dependent insulinotropic polypeptide (GIP) is an incretin secreted during the intake of a meal by the gastrointestinal tract and stimulates pancreatic β-cells to secrete insulin. GIP is secreted from the K-cells of the small intestine and exerts its effects through a specific receptor, the GIP receptor (GIPR) [11,12]. GIPR is expressed in many other organs, such as the nervous system, adipose tissue, heart, lung, bone, and immune cells, and exhibits extra-pancreatic effects [13,14,15,16]. Mice lacking GIPR (GIPRKO) demonstrated lower bone formation [17]. Bone histomorphometric analyses have revealed that the trabeculae tend to be thinner in GIPRKO, and cellular activity parameters of bone formation have revealed that bone formation parameters in GIPRKO are significantly lower than in *GIPR*^+/+^ mice. However, the results regarding the cellular activity parameters of bone resorption in GIP are controversial. Zhong et al. found that GIP inhibited bone resorption in an organ culture system and resorptive activity of mature osteoclasts [18]. Tsukiyama et al. demonstrated that GIP did not inhibit the pit-forming activity of osteoclasts [17].

To reveal the impacts of GIP on bone remodeling, we performed orthodontic tooth movements by attaching a closed-coil spring between the maxillary incisors and the left first molar. Bone remodeling can be demonstrated by a constant and continuous force using an orthodontic tooth movement system. Using this method, we investigated the impact of GIP on traction force-induced bone remodeling in the GIPRKO and WT.

## 2. Results

### 2.1. Animal Characteristics

The body weight and blood glucose levels of 8-week-old male WT (*n* = 5) and GIPRKO (*n* = 5) at the time of coil-spring installation were assessed (Figure 1A). No significant difference was observed in the body weights and blood glucose levels of the GIPRKO compared with the WT (Figure 1B,C).

### 2.2. Evaluation of Orthodontic Tooth Movement

Figure 2A,B show a 10-gf Ni-Ti closed-coil spring which was attached to the maxillary incisors and the maxillary left first molars (M1) of 8-week-old male WT and GIPRKO. Orthodontic tooth movement was induced by 10 gf traction using this closed-coil spring. Fourteen days after coil installation, the maxillae of the WT and GIPRKO were analyzed by μCT. Since the coil spring was dropped in one mouse of the GIPRKO, it was excluded from the subsequent analyses. Figure 3A,B show μCT images which were obtained in both sagittal and horizontal directions. When the distance between M1 and M2 was measured for analysis using TRI / 3D-BON, the distances of the tooth movement by coil load in the GIPRKO was significantly increased by 2.0-fold compared with the movements achieved in the WT (WT 112.0 ± 13.1 μm, GIPRKO 226.0 ± 74.0 μm, *p <* 0.05) (Figure 3C).

### 2.3. Residual Alveolar Bone Amount and Trabecular Structure

Figure 4A shows a pattern diagram and a calculation method of the alveolar bone in the inter-root septum from the root bifurcation to the apex of M1, measured 14 days after the orthodontic tooth movement (Figure 4A). Both WT and GIPRKO exhibited significantly decreased amounts of residual alveolar bone on the loaded side compared with those on the control side (WT loaded: 21.7%; GIPRKO loaded: 34.8%) (*p <* 0.01, *p <* 0.001, respectively) (Figure 4B). A significant decrease of 25.1% in the residual alveolar bone amount was observed on the loaded side of the GIPRKO compared with in the WT (*p <* 0.05). 

Subsequently, we analyzed the trabecular structures at the same site. Tb.Th. was significantly decreased on the loaded side compared with the control side by 14.0% and 14.8% in WT and GIPRKO, respectively (*p <* 0.01, *p <* 0.001, respectively) (Figure 5A). In Tb.N., the WT did not show a significant difference between the control and the loaded sides, whereas the GIPRKO showed a significant decrease on the loaded side (14.4%) compared with the control side (*p <* 0.001) (Figure 5B). There was a significant reduction in Tb.N.

On the loaded side of the GIPRKO compared with the loaded side of the WT (14.3% decrease, *p <* 0.01). In Tb.Sp., the WT did not show a significant difference between the control and the loaded sides (Figure 5C). The GIPRKO showed a significant increase in the Tb.Sp. on the loaded side (150.9%) compared with the control side (*p <* 0.001). A significant increase in Tb.Sp. was observed (132.3%) on the loaded side of the GIPRKO compared with the WT (*p <* 0.01).

### 2.4. Osteoclast Formation by Orthodontic Tooth Movement

Histological observations revealed the enlargement of the periodontal ligament cavity around M1 on the traction side, narrowing on the compression side, as well as alveolar bone resorption. Figure 6 shows osteoclasts that were visualized by TRAP staining on the M1 compression side of the WT and GIPRKO 14 days after coil loading. In both the WT and GIPRKO, the number of osteoclasts was significantly increased on the loaded side compared with the control side by 5.7-fold and 7.0-fold, respectively (*p <* 0.001) (Figure 6B). The number of osteoclasts increased 1.7-fold on the loaded side of the GIPRKO compared with the WT (*p <* 0.05).

### 2.5. Osteoblast Formation by Orthodontic Tooth Movement

Osteoblasts were analyzed by ALP staining. Figure 7 shows the osteoblasts on the M1 traction side of the WT and GIPRKO 14 days after orthodontic tooth movement. The number of osteoblasts on the alveolar bone in the control side of the GIPRKO was significantly decreased by 19.2% compared with the control side of the WT (*p <* 0.05). In the WT, orthodontic tooth movement significantly increased the number of osteoblasts round the individual teeth by 1.2-fold compared with the control side, whereas the GIPRKO showed no such increase.

## 3. Discussion

We investigated the effects of GIP on force-induced bone remodeling. Orthodontic tooth movement by 10 gf traction using a Ni-Ti closed-coil spring demonstrated that the distance of tooth movement was 2-fold faster in the GIPRKO than in the WT. Analyses of the trabecular structure revealed that the GIPRKO expressed decreased Tb.N. and increased Tb.Sp. following orthodontic tooth movements. The GIPRKO exhibited no increase in the number of osteoblasts by orthodontic tooth movement; however, the number of osteoclasts on the coil-loaded side was significantly increased in the GIPRKO compared with the WT.

Orthodontic tooth movement is orchestrated by osteoclastogenesis and osteogenesis and also is a process that combines both pathologic and physiologic responses to externally applied forces [19]. In the orthodontic tooth movement, inflammation occurs initially at compression sites, which is caused by the constriction of the periodontal ligament, microvasculature, resulting in a focal necrosis [20]. Osteoclasts are recruited from the adjacent marrow spaces in the compression site [21]. Osteoblast differentiation is an important process on the tension side during orthodontic tooth movement. The initial response to orthodontic tooth movement on the tension side is a proliferation of osteoblast progenitors [22,23]. However, the mechanisms of osteogenesis at the tension sites are not well documented. 

GIPR is expressed both in osteoblasts and osteoclasts [18,24,25]. GIP directly increases in lysyl oxidase activity and collagen maturity by activating the adenylyl cyclase–cAMP pathway in MC3T3-E1 cells that are phenotypically normal osteoblasts [26]. It is also reported that GIP suppresses osteoblast apoptosis [17] On the other hand, the effects of GIP on osteoclasts are controversial. Zhong et al. found that GIP inhibited bone resorption in an organ culture system and resorptive activity of mature osteoclasts [18]. Tsukiyama et al. demonstrated that GIP did not inhibit the pit-forming activity of osteoclasts [17]. There is no difference of longitudinal growth of limb bones between GIPR^−/−^ mice and GIPR^+/+^ mice. However, histological analyses of tibiae in trabeculae tended to be thinner in GIPR^−/−^ mice.

In this study, the number of osteoblasts on the control side was decreased in the GIPRKO compared with the WT, whereas the number of osteoclasts on the control side did not change between the GIPRKO and WT. Our results suggest that the deletion of GIP signaling may suppress the formation of osteoblasts but does not affect osteoclasts in the steady state. Both osteoblasts and osteoclasts increased around the coil-loaded tooth in the WT. However, the osteoblasts in the GIPRKO did not increase around the coil-loaded tooth, indicating that GIP signaling plays a critical role in the formation of osteoblasts, both in the steady state and in force-induced bone remodeling. Although there was no significant difference in the formation of osteoclasts in the steady state between the WT and the GIPRKO, the GIPRKO showed significant greater osteoclast formation during bone remodeling by traction force. 

Analyses of the trabecular structure revealed that 10 gf traction induced a decrease in the trabecular width in both the WT and GIPRKO. The GIPRKO showed a significantly decreased number of trabeculae and increased trabecular space with 10 gf traction, but the WT did not. These results indicate a pivotal role for GIP in bone remodeling.

The direction of force exerted by the coil-spring was maintained at the level of the occlusal plane over 12 weeks [27]. This model allows us to address a constant force to alveolar bone via M1. Orthodontic tooth movement models can be used not only for the study of orthodontic treatment but also for the dynamics of bone remodeling. Using this model, we successfully demonstrated that GIP increases osteoblast formation in both steady state and in force-induced bone remodeling and suppresses the formation of osteoclasts in force-induced bone remodeling. 

With an increased number of adults seeking orthodontic treatments, the number of patients with a high risk of osteoporosis who seek orthodontic treatments are increasing. GIP would be useful for these patients in orthodontic treatments. Furthermore, since osteoporosis itself is a disturbance of bone remodeling, GIP may be a novel treatment for osteoporosis. Further study is required on this issue. 

## 4. Materials and Methods

### 4.1. Animals

GIPRKO were generated as previously described [23]. Male C57BL/C mice (WT) were obtained from CLEA Japan (Tokyo, Japan). All mice were housed in individual cages under controlled temperature (22 ± 2.0 °C) with a humidity of 50 ± 10%. The feed was CE2 type powder feed (CLEA Japan, Tokyo, Japan) and tap water was used as drinking water, both of which were freely ingested. The Institutional Animal Care and Use Committee of Aichi Gakuin University approved all experimental protocols (ethical clearance number: AGUD 381).

### 4.2. Orthodontic Tooth Movement

The 8-week-old male GIPRKO and WT were used in the present study. Under general anesthesia by intraperitoneal administration of three mixed anesthetics: medetomidine hydrochloride (Meiji Seika Pharma, Tokyo, Japan), midazolam (Astellas Pharma, Tokyo, Japan), and butorphanol tartrate (Meiji Seika Pharma, Tokyo, Japan), a 10 gf Ni-Ti closed-coil spring was attached between the maxillary first molar and the left first molar, causing mesial movement of the left first molar (*n* = 5 from each group). The spring delivers a constant and continuous force of approximately 10 gf [19]. M1 on the right side was used as a control. The following experiments were conducted two weeks after the installation of the coil spring.

### 4.3. Microcomputer Tomography

Maxillary bone correction 14 days after orthodontic tooth movement was analyzed using a microcomputed tomography (μCT) device (Rigaku, Tokyo, Japan). The tube voltage, current, imaging time, and pixel size were 90 kV, 150 μA, 2 min, and 20, respectively. Three-dimensional analysis was performed at 20 × 20 μm. The tooth movement distance was performed using the software TRI/3D-BON (Ratoc System Engineering, Osaka, Japan). The movement distance of the maxillary teeth was measured by adjusting the sagittal and horizontal sections so that restenosis could be observed. The image was rotated and adjusted to ensure that an occlusal view with the narrowest gap between M1 and M2 was observed. To observe the condition of the alveolar crest of the left M1 alveolar septum, the root of M1 and surrounding alveolar bone were observed in a horizontal plane. The bone volume (BV) was normalized to the total volume of the sample (TV) in the inter-root septum to the relative bone volume (BV/TV) according to the method described by Sprogar et al. [28]. The mean trabecular thickness (Tb.Th.) was determined from the local thickness at each bone. The trabecular separation (Tb.Sp.) was calculated using the direct thickness calculation of the non-bone sections of the 3D image.

### 4.4. Histopathological Observation

The maxilla was harvested 14 days after orthodontic tooth movement and fixed in a 10% neutral buffered formalin solution. Next, it was decalcified with 10% EDTA (pH 7.2) for approximately 4 weeks at 4°C and paraffin-embedded according to the conventional method to prepare a continuous tissue section of 5 μm in the horizontal section direction. The sections were stained with tartrate-resistant acid phosphatase (TRAP) and alkaline phosphatase (ALP) (September Sapie Co., Ltd., Tokyo, Japan) to observe osteoclasts and osteoblasts. Osteoclasts were defined as polynuclear TRAP-positive cells that were in contact with the alveolar bone. The periodontal tissue was observed under an optical microscope. One-third of the root bifurcation, divided into three equal parts from the root bifurcation to the apex, was used as the tissue observation site. Regarding the measurement of the number of osteoclasts after TRAP staining, the circumference of the alveolar bone surface on the compression side around the distal palatal root (M1DP) of the first maxillary molar was measured and the osteoclasts on the alveolar bone surface were measured. The number of samples was measured. For the measurement of osteoblasts, the circumference of the alveolar bone surface on the traction side around the mesial root (M1M) of the maxillary first molar was measured and the number of ALP-positive cells per length of the alveolar bone surface was calculated. 

### 4.5. Statistical Analyses 

Data were processed using GraphPad Prism (GraphPad Software, San Diego, CA, USA) and are presented as the mean ± SEM. Statistical significance was analyzed using one-way ANOVA and Bonferroni correction for multiple comparisons. Differences with *p* < 0.05 were considered significant.

## 5. Conclusions

In conclusion, we have demonstrated the role of GIP in the dynamics of bone remodeling. We revealed that GIP played a role in the maintenance of osteoblasts at the steady state and increased osteoblasts in traction force-induced bone remodeling. On the contrary, GIP suppressed the number of osteoclast formations in the traction force-induced bone remodeling, although GIP did not affect the number of osteoclasts in the steady state.

## Figures and Tables

**Figure 1 ijms-23-08922-f001:**
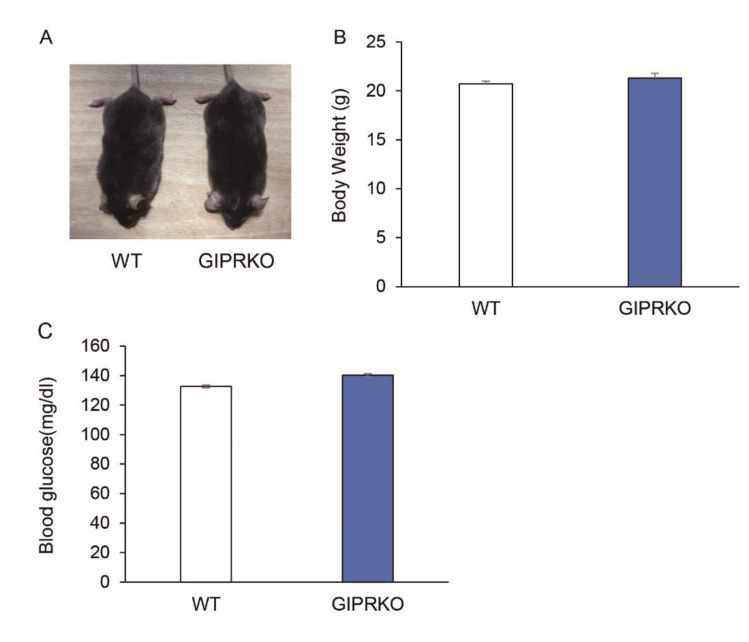
Body weights and blood glucose levels of WT and GIPRKO. (**A**) Photos of mice from the WT and the GIPRKO. (**B**) Body weights (*n* = 5) and (**C**) blood glucose levels (*n* = 5) were represented. Results are expressed as mean ± SEM.

**Figure 2 ijms-23-08922-f002:**
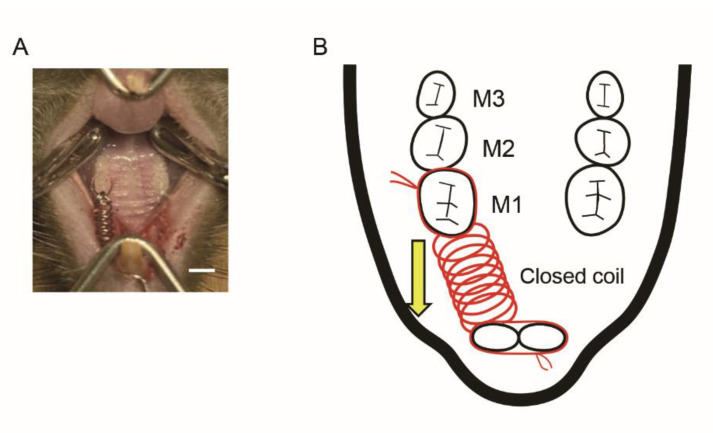
(**A**) A 10-gf Ni-Ti closed-coil spring was attached to the maxillary incisors and the maxillary left first molars (M1) of 8-week-old male WT and GIPRKO, and orthodontic tooth movement was administered for 14 days. Scale bar: 2.0 mm. (**B**) Schematic diagram of orthodontic tooth moving device.

**Figure 3 ijms-23-08922-f003:**
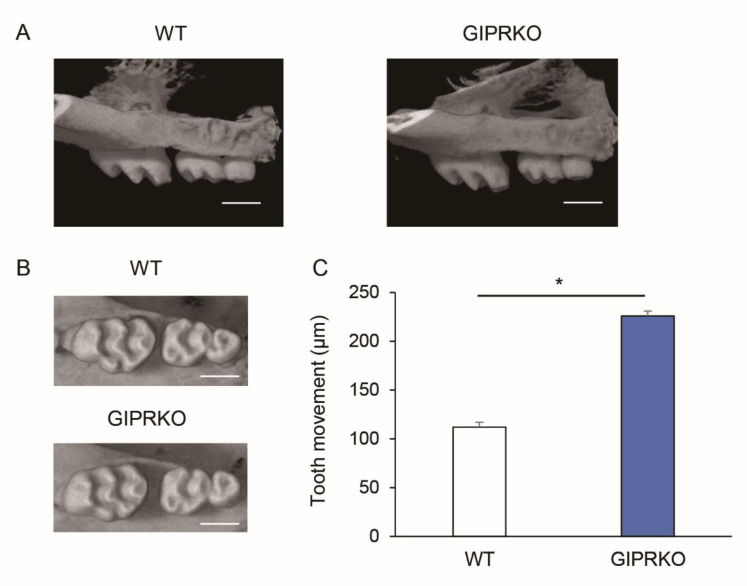
Orthodontic tooth movement in the WT and the GIPRKO. Comparison of μCT images 14 days after orthodontic tooth movement. (**A**) Sagittal section and (**B**) horizontal section. Scale bar: 1.0 mm. (**C**) Average distance between M1 and M2 14 days after orthodontic tooth movement (* *p <* 0.05) (WT: *n* = 5, GIPRKO: *n* = 4).

**Figure 4 ijms-23-08922-f004:**
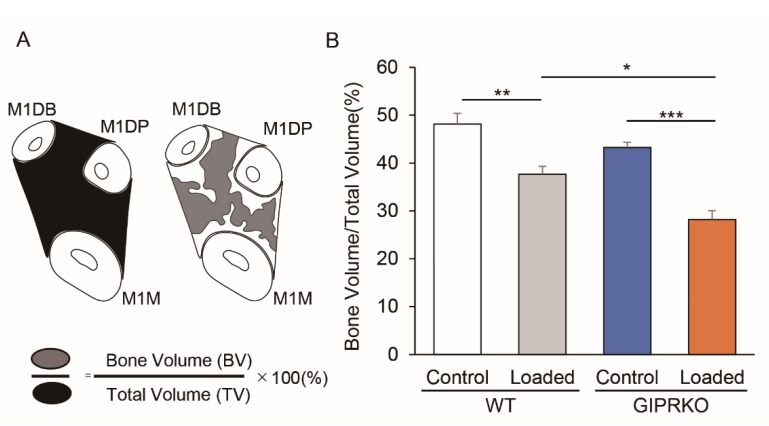
Alveolar bone in the inter-root septum from the root bifurcation to the apex of M1 was measured 14 days after the orthodontic tooth movement. (**A**) Schematic diagram of the horizontal cross section of the interroot septum of M1. (**B**) Comparison of bone volume/total volume (BV/TV) between M1M: M1 mesial root, M1DP: M1 centrifugal palatal root, M1DB: M1 centrifugal buccal root in the control and the loaded side in WT and GIPRKO (* *p <* 0.05, ** *p <* 0.01, *** *p <* 0.001) (WT: *n* = 5, GIPRKO: *n* = 4).

**Figure 5 ijms-23-08922-f005:**
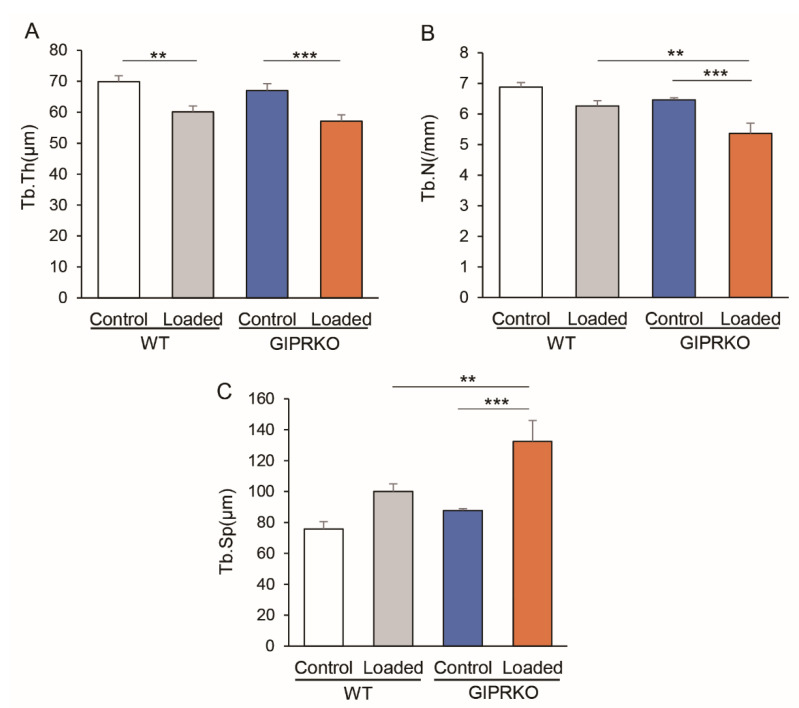
Measurement of trabecular structure of WT and GIPRKO. The mean trabecular thickness (Tb.Th.) was determined from the local thickness at each representative bone. The trabecular separation (Tb.Sp.) was calculated by calculating the direct thickness to the non-bone sections of the 3D image. (**A**) Comparison of Tb.Th. of the M1 inter-root septum (** *p* < 0.01, *** *p* < 0.001). (**B**) Comparison of the number of trabeculae (** *p <* 0.01, *** *p <* 0.001). (**C**) Comparison of the Tb.Sp. (** *p <* 0.01, *** *p <* 0.001) (WT: *n* = 5, GIPRKO: *n* = 4).

**Figure 6 ijms-23-08922-f006:**
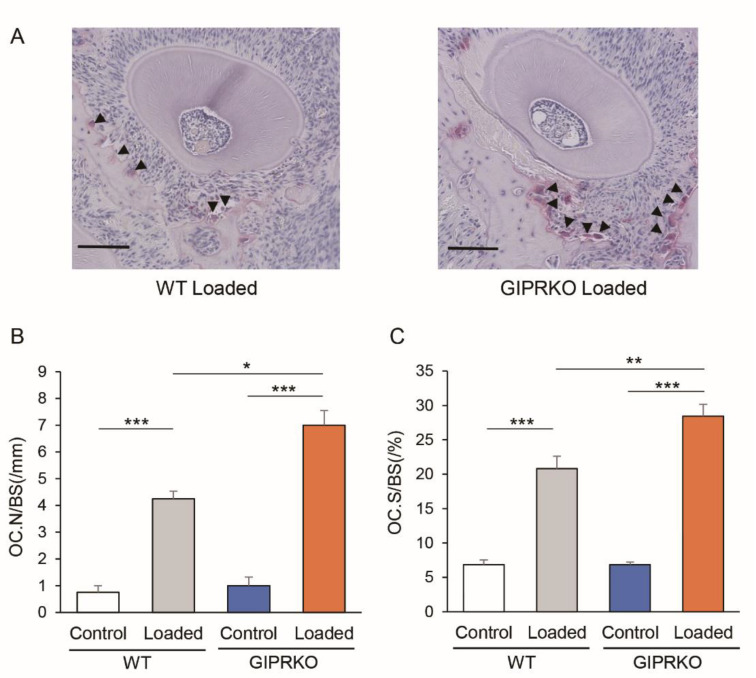
Evaluation of osteoclasts by TRAP staining. Comparison of the number of polynuclear TRAP-positive cells on the compression side of M1DP. (**A**) TRAP-stained micrographs of the experimental side of the WT and GIPRKO. Black arrow head points to osteoclast. (Scale bar: 100 μm. (**B**) Comparison of changes in the osteoclast number (OC.N) and bone surface (BS). (* *p <* 0.05, *** *p <* 0.001). (**C**) Comparison of changes in the osteoclast surface (OC.S) and BS (** *p <* 0.01, *** *p <* 0.001) (WT: *n* = 5, GIPRKO: *n* = 4).

**Figure 7 ijms-23-08922-f007:**
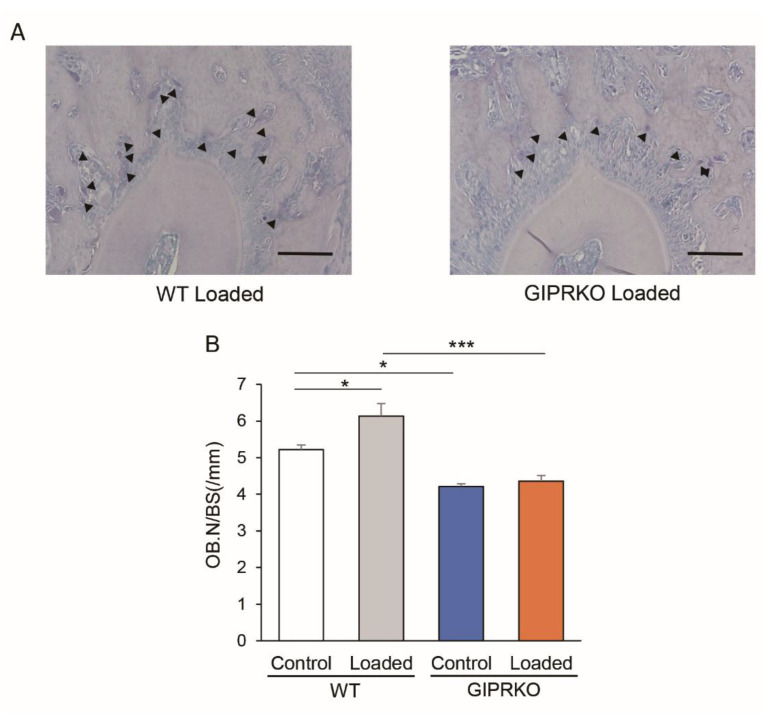
Evaluation of osteoblasts by ALP staining. Comparison of ALP-positive cells on the traction side of M1M. (**A**) ALP-stained micrographs of the loaded side of the WT and GIPRKO. Black arrow head points to osteoblast. Scale bar: 100 μm. (**B**) Comparison of changes in the osteoblast number per BS (OB.N./BS). (* *p <* 0.05, *** *p <* 0.001). (WT: *n* = 5, GIPRKO: *n* = 4).

## Data Availability

The data presented in this study are available on request form the corresponding author.

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
