# Peer review of "Impacts of Glucose-Dependent Insulinotropic Polypeptide on Orthodontic Tooth Movement-Induced Bone Remodeling"

_ijms, 2022, doi:10.3390/ijms23168922_

Round 1
Reviewer 1 Report
Manuscript Number: 1843093
Title:
Ok.
Abstract:
Generally ok.
Introduction:
Line 42: The sentence can be improved. What do you mean by ‘ have been reported?
Line 66 to 75: Much of this paragraph is about the outcome of your research. This should be improved i.e. only highlight study aims.
Results:
Line 78: Please improve the sentence. Figure 1A shows….Please do the same for the rest. This sounds better.
Line 80: …compared with WT.
Line 89: Please use small p for p-value. Please change throughout the text.
Discussion:
Line 175 to 191: This section is more like explaining the results. Please improve section by adding more supporting evidence.
Methodology:
- Line 225: Is AGUD 381 the ethical clearance number? If yes, please add more detail i.e (Ethical clearance number: AGUD 381).
- Line 240: The company name has to be italicized?
Conclusion:
Ok. Reflecting the content.
Author Response
Responses to the comments of Reviewer #1
We thank the reviewer for the time and efforts spent for our manuscript and thoughtful comments for the improvement of our manuscript. We herein respond to your suggestions.
#1. Intrudction Line 42 The sentence can be improved. What do you mean by ‘ have been reported?
Thank you for your comment.
We revised our manuscript as follows;
“In recent years, middle-aged and elderly patients who desire orthodontic treatment are increased”
#2. Introduction Line 66-75 Much of this paragraph is about the outcome of your research. This should be improved i.e. only highlight study aims.
Thank you for your valuable comment. We revised these sentences to highlight study aims.
The following sentences were revised in the revised manuscript;
(Line 66-70)
“To reveal the impacts of GIP on bone remodeling, we performed orthodontic tooth movements by attaching closed-coil spring between the maxillary incisors and the left first molar. Bone remodeling can be demonstrated by a constant and continuous force using an orthodontic tooth movement system. Using this method, we investigated the impact of GIP on traction force-induced bone remodeling in the GIPRKO and WT.”
#3. Results: Line 78 Please improve the sentence. Figure 1A shows….Please do the same for the rest. This sounds better.
Thank you for your valuable comments. We revised throughout our manuscript according to the reviewer’s comment.
#4. Results: Line 80 …compared with WT.
We revised our manuscript.
#5. Results: Line 89Please use small p for p-value. Please change throughout the text.
We changed to small p for p-value throughout the text.
#6. Discussion: Line 175 to 191: This section is more like explaining the results. Please improve section by adding more supporting evidence.
We summarized the results compactly and discussed it with the evidences.
The following sentences are revised in discussion:
(Line 174-211)
“We investigated the effects of GIP on force-induced bone remodeling. Orthodontic tooth movement by 10 gf traction using a Ni-Ti closed-coil spring demonstrated that the distance of tooth movement was 2-fold faster in the GIPRKO than in the WT. Analyses of the trabecular structure revealed that the GIPRKO expressed decreased Tb.N. and increased Tb.Sp. following orthodontic tooth movements. The GIPRKO exhibited no increase in the number of osteoblasts; however, the number of osteoclasts on the coil-loaded side was significantly increased in the GIPRKO compared to that in the WT.
Orthodontic tooth movement is orchestrated by osteoclastogenesis and osteogenesis and also is a process that combines both pathologic and physiologic responses to externally applied forces [19]. In the orthodontic tooth movement, inflammation is occurred initially at compression sites, which is caused by constriction of the periodontal ligament, microvasculature, resulting in a focal necrosis [20]. Osteoclasts are recruited from the adjacent marrow spaces in the compression site [21]. Osteoblast differentiation is an important process on the tension side during orthodontic tooth movement. The initial response to orthodontic tooth movenent on the tension side is a proliferation of osteoblast progenitors [22, 23]. However, the mechanisms of osteogenesis at the tension sites are not well documented.
GIPR is expressed both in osteoblasts and osteoclasts [18, 24, 25]. GIP directly increases in lysyl oxidase activity and collagen maturity in MC3T3-E1 cells which are phenotypically normal osteoblasts [26]. GIP suppresses osteoblast apoptosis and increases lysyl oxidase activity and collagen maturity by activating the adenylyl cyclase–cAMP pathway [17, 26] On the other hand, the effects of GIP on osteoclasts are controversial. Zhong et al. found that GIP inhibited bone resorption in an organ culture system and resorptive activity of mature osteoclasts [18]. Tsukiyama et al. demonstrated that GIP did not inhibit the pit-forming activity of osteoclasts [17]. There is no difference of longitudinal growh of limb bones between GIPR-/- mice and GIPR+/+ mice. However, histological analyses of tibiae in trabeculae tended to be thinner in GIPR-/- mice[17].
In this study, the number of osteoblasts on the control side was decreased in the GIPRKO compared to that in the WT, whereas the number of osteoclasts did not change between the GIPRKO and WT. Our results suggest that the deletion of GIP signaling may suppress the formation of osteoblasts but does not affect osteoclasts in the steady state. Both osteoblasts and osteoclasts increased around the coil-loaded tooth in the WT. However, the osteoblasts in the GIPRKO did not increase around the coil-loaded tooth, indicating that GIP signaling plays a critical role in the formation of osteoblasts, both in the steady state and in force-induced bone remodeling. Although there was no significant difference in the formation of osteoclasts in the steady state between the WT and GIPRKO, the GIPRKO showed significant greater osteoclast formation during bone remodeling by traction force. ”
#7. Methodology: Line 225: Is AGUD 381 the ethical clearance number? If yes, please add more detail i.e (Ethical clearance number: AGUD 381).
We revised our manuscript as the reviewer mentioned.
#8. Methodology: -Line 240: The company name has to be italicized?
I do not think that we need to be italicizied for company names. I changed the company name to capitalize only the begging of words.

Reviewer 2 Report
The current study used GIP-R KO mice to demonstrate the effect of GIP on the dynamics of bone remodeling. I'd like to make the following observations.
Major Concerns:
1. Functional differences between GIPR-KO mice and WT mice may lead to a better understanding of GIP's role. However, there is some bias involved. During the development of KO mice, new homeostasis will emerge to compensate for the loss of GIP function. Integration of endogenous peptides with GIP may participate in KO mice to result in the current findings. How can this worry be dispelled?
2. Qualified osteoclasts were identified using TRAP staining. Osteoclast quantification is widely used for more reliable results.
3. There was no application of current findings in clinical practice.
Minor Points:
1. GIP-origin KO's Mice appears perplexed. The reference(s) for Orthodontic Tooth Movement are required. Both belonged to the focal point of the current study.
2. GIPRKO's typo must be corrected.
3. This orthodontic tooth movement system, which also requires reference, can demonstrate bone remodeling (s).
4. The sample size must be clearly visible in all figures. The old method (n = 4-5) was no longer used.
5. It denotes unclear that the data in all figures were obtained from the same mice. Please be as specific as possible.
6. The role of GIP in bone pathophysiology must be thoroughly discussed.
Author Response
dResponses to the comments of Reviewer #2
We thank the reviewer for the time and efforts spent for our manuscript and thoughtful comments for the improvement of our manuscript. We herein respond to your suggestions.
#1. Functional differences between GIPR-KO mice and WT mice may lead to a better understanding of GIP's role. However, there is some bias involved. During the development of KO mice, new homeostasis will emerge to compensate for the loss of GIP function. Integration of endogenous peptides with GIP may participate in KO mice to result in the current findings. How can this worry be dispelled?
We can not deny the possibility that there may be new homeostasis will emerge to compensate for the loss of GIP function. However, as we showed in figure 1, there were no significant differences between the weights and the blood glucose levels in GIPRKO and WT, which was consistent with the previous study of GIPRKO[1]. Although the baseline blood glucose level and insulin level were not different, GIPRKO showed higher blood glucose levels with impaired initial insulin response compared with age-matched age-matched GIPR+/+ mice in the oral glucose tolerance test. Since GIP increases by oral glucose intake and leads to secret insulin, these phenomena in GIPRKO indicate the lack of GIP signal in GIPRKO.
- Miyawaki, K.; Yamada, Y.; Yano, H.; Niwa, H.; Ban, N.; Ihara, Y.; Kubota, A.; Fujimoto, S.; Kajikawa, M.; Kuroe, A.; Tsuda, K.; Hashimoto, H.; Yamashita, T.; Jomori, T.; Tashiro, F.; Miyazaki, J.; Seino, Y., Glucose intolerance caused by a defect in the entero-insular axis: a study in gastric inhibitory polypeptide receptor knockout mice. Proc Natl Acad Sci U S A 1999, 96, 14843-14847.
#2. Qualified osteoclasts were identified using TRAP staining. Osteoclast quantification is widely used for more reliable results.
We investigated the impact of GIP on traction forth-induced bone remodeling in vivo. Gene expressions of markers of osteoclastic differentiation, such as c-fms, TRAP, cathepsin K, are used to osteoclast quantification. However, since we investigated the model of orthodontic tooth movement, there are both the compression side and the traction side around the M1, we could not separate the tissue exactly. For this reason, we used TRAP staining to detect osteoclasts in vivo.
#3. There was no application of current findings in clinical practice.
Our findings indicate that GIP increases osteoblast formation in both steady state and in force-induced bone remodeling and suppresses the formation of osteoclasts in force-induced bone remodeling. With increased adults seeking orthodontic treatments, the number of patients with the high risk of osteoporosis who seek orthodontic treatments are increasing. GIP would be useful for these patients in orthodontic treatments. Furthermore, osteoporosis itself is a disturbance of bone remodeling, GIP may be a novel treatment for osteoporosis. Further study requires in this issue.
Following sentences are inserted in the revised manuscript.
(Line 223-227)
“With increased adults seeking orthodontic treatments, the number of patients with the high risk of osteoporosis who seek orthodontic treatments are increasing. GIP would be useful for these patients in orthodontic treatments. Furthermore, since osteoporosis itself is a disturbance of bone remodeling, GIP may be a novel treatment for osteoporosis. Further study requires in this issue.”
#4. GIP-origin KO's Mice appears perplexed. The reference(s) for Orthodontic Tooth Movement are required. Both belonged to the focal point of the current study.
We discussed what the reviewer mentioned as follows in Discussion:
(Line 181-211)
“Orthodontic tooth movement is orchestrated by osteoclastogenesis and osteogenesis and also is a process that combines both pathologic and physiologic responses to externally applied forces [19]. In the orthodontic tooth movement, inflammation is occurred initially at compression sites, which is caused by constriction of the periodontal ligament, microvasculature, resulting in a focal necrosis [20]. Osteoclasts are recruited from the adjacent marrow spaces in the compression site [21]. Osteoblast differentiation is an important process on the tension side during orthodontic tooth movement. The initial response to orthodontic tooth movenent on the tension side is a proliferation of osteoblast progenitors [22, 23]. However, the mechanisms of osteogenesis at the tension sites are not well documented.
GIPR is expressed both in osteoblasts and osteoclasts [18, 24, 25]. GIP directly increases in lysyl oxidase activity and collagen maturity in MC3T3-E1 cells which are phenotypically normal osteoblasts [26]. GIP suppresses osteoblast apoptosis and increases lysyl oxidase activity and collagen maturity by activating the adenylyl cyclase–cAMP pathway [17, 26] On the other hand, the effects of GIP on osteoclasts are controversial. Zhong et al. found that GIP inhibited bone resorption in an organ culture system and resorptive activity of mature osteoclasts [18]. Tsukiyama et al. demonstrated that GIP did not inhibit the pit-forming activity of osteoclasts [17]. There is no difference of longitudinal growh of limb bones between GIPR-/- mice and GIPR+/+ mice. However, histological analyses of tibiae in trabeculae tended to be thinner in GIPR-/- mice[17].
In this study, the number of osteoblasts on the control side was decreased in the GIPRKO compared to that in the WT, whereas the number of osteoclasts did not change between the GIPRKO and WT. Our results suggest that the deletion of GIP signaling may suppress the formation of osteoblasts but does not affect osteoclasts in the steady state. Both osteoblasts and osteoclasts increased around the coil-loaded tooth in the WT. However, the osteoblasts in the GIPRKO did not increase around the coil-loaded tooth, indicating that GIP signaling plays a critical role in the formation of osteoblasts, both in the steady state and in force-induced bone remodeling. Although there was no significant difference in the formation of osteoclasts in the steady state between the WT and GIPRKO, the GIPRKO showed significant greater osteoclast formation during bone remodeling by traction force.”
#5. GIPRKO's typo must be corrected.
Thank you for your comment. We corrected the mistype.
#6. This orthodontic tooth movement system, which also requires reference, can demonstrate bone remodeling (s).
We discussed what the reviewer mentioned as follows:
(Line181-190)
“Orthodontic tooth movement is orchestrated by osteoclastogenesis and osteogenesis and also is a process that combines both pathologic and physiologic responses to externally applied forces [19]. In the orthodontic tooth movement, inflammation is occurred initially at compression sites, which is caused by constriction of the periodontal ligament, microvasculature, resulting in a focal necrosis [20]. Osteoclasts are recruited from the adjacent marrow spaces in the compression site [21]. Osteoblast differentiation is an important process on the tension side during orthodontic tooth movement. The initial response to orthodontic tooth movenent on the tension side is a proliferation of osteoblast progenitors [22, 23]. However, the mechanisms of osteogenesis at the tension sites are not well documented.”
#7. The sample size must be clearly visible in all figures. The old method (n = 4-5) was no longer used.
We revised our manuscript according to the reviewer’s comment.
#8. It denotes unclear that the data in all figures were obtained from the same mice. Please be as specific as possible.
We used five WT mice and five GIPRKO in this study. Since the coil spring was dropped in one mice of GIPRKO, it was excluded from the subsequent analyses. The data from figure 3 to figure 7 were obtained the same mice (WT: n=5, GIPRKO: n=4).
#9. The role of GIP in bone pathophysiology must be thoroughly discussed.
We discussed what the reviewer mentioned as follows:
(Line 191-211)
“GIPR is expressed both in osteoblasts and osteoclasts [18, 24, 25]. GIP directly increases in lysyl oxidase activity and collagen maturity in MC3T3-E1 cells which are phenotypically normal osteoblasts [26]. GIP suppresses osteoblast apoptosis and increases lysyl oxidase activity and collagen maturity by activating the adenylyl cyclase–cAMP pathway [17, 26] On the other hand, the effects of GIP on osteoclasts are controversial. Zhong et al. found that GIP inhibited bone resorption in an organ culture system and resorptive activity of mature osteoclasts [18]. Tsukiyama et al. demonstrated that GIP did not inhibit the pit-forming activity of osteoclasts [17]. There is no difference of longitudinal growh of limb bones between GIPR-/- mice and GIPR+/+ mice. However, histological analyses of tibiae in trabeculae tended to be thinner in GIPR-/- mice[17].
In this study, the number of osteoblasts on the control side was decreased in the GIPRKO compared to that in the WT, whereas the number of osteoclasts did not change between the GIPRKO and WT. Our results suggest that the deletion of GIP signaling may suppress the formation of osteoblasts but does not affect osteoclasts in the steady state. Both osteoblasts and osteoclasts increased around the coil-loaded tooth in the WT. However, the osteoblasts in the GIPRKO did not increase around the coil-loaded tooth, indicating that GIP signaling plays a critical role in the formation of osteoblasts, both in the steady state and in force-induced bone remodeling. Although there was no significant difference in the formation of osteoclasts in the steady state between the WT and GIPRKO, the GIPRKO showed significant greater osteoclast formation during bone remodeling by traction force.”
Round 2
Reviewer 2 Report
It has been improved in a good way.